# Validation of FES-I and Short FES-I Scales in the Polish Setting as the Research Tools of Choice to Identify the Fear of Falling in Older Adults

**DOI:** 10.3390/ijerph192416907

**Published:** 2022-12-16

**Authors:** Marek Zak, Marta Makara-Studzińska, Agnieszka Mesterhazy, Jacek Mesterhazy, Paweł Jagielski, Aneta Januszko-Szakiel, Tomasz Sikorski, Piotr Jaworski, Renata Miszczuk, Waldemar Brola

**Affiliations:** 1Institute of Health Sciences, Collegium Medicum, Jan Kochanowski University, Ul. Zeromskiego 5, 25-369 Kielce, Poland; 2Institute of Nursing and Midwifery, Department of Health Psychology, Faculty of Health Sciences, Collegium Medicum, Jagiellonian University, Ul. Kopernika 25, 31-501 Krakow, Poland; 3Department of Nutrition and Drug Research, Faculty of Health Sciences, Collegium Medicum, Jagiellonian University, Skawinska 8, 31-066 Krakow, Poland; 4Institute of Information Studies, Faculty of Managment and Social Communication, Jagiellonian University, Ul. Lojasiewicza 4, 30-348 Krakow, Poland; 5Institute of Pedagogy, Jan Kochanowski University, Ul. Zeromskiego 5, 25-369 Kielce, Poland

**Keywords:** FES, validation, fear of falling, older adults

## Abstract

Fear of falling is associated with a clear hazard to individual self-reliance, reduced physical activity, as well as a sense of shame and loss of self-confidence. The present study aimed to complete the applicable translation and validation protocol for the Falls Efficacy Scale—International (FES-I) tool, following its prior adaptation to ensure full compatibility with the Polish setting. The FES-I questionnaire, along with its abridged version, was translated in line with the recommended standards of the MAPI Institute, taking into account both the cultural fabric and pertinent language specifics of the country. The survey was attended by 740 individuals (N = 740; 463 women, 277 men), over 60 years old. All respondents were required to complete both the FES-I and FES-I (Short) questionnaires twice, following an intervening period, and subsequently had their responses statistically assessed. The FES-I questionnaire, along with its abridged version, may be recommended as an effective assessment tool for addressing the fear of falling issue among the older adults, consequently allowing the teams of attending physicians, physiotherapists, psychologists, or psychiatrists to complete an unambiguous diagnosis, with a view to helping the patients overcome this particular type of anxiety.

## 1. Introduction

One of the more common and serious health problems affecting the elderly population is posed by accidental falls [1,2,3]. Ageing of populations is associated with numerous social, psychological, economic, and political consequences, whereas a group of older adults over 65 years of age falls well within the majority of health care recipients [4,5,6]. Incidence of falls and locomotion disorders, dementia syndromes, depression, visual and hearing impairments, incontinence, multi-medication and multi-disease, and geriatric iatrogenic syndrome make up the so-called Geriatric Giants [7,8,9,10]. Older adults suffer from chronic, complex disorders which may consequently deteriorate into disability, whilst at the same time being instrumental in overall loss of self-reliance, appreciably reduced scope in the activities of daily living, and a diminished overall quality of life [7,11,12,13]. 

Fear of falling (FoF) is a specific type of anxiety associated with both the physical consequences of sustaining an incidental fall, such as gait disturbance, impaired balance, disability, and a variety of other adverse consequences in social life, such as a sense of shame, reduced self-reliance, and loss of self-confidence [14,15,16]. As per a body of research conducted to date, FoF in older adults living on their own has been estimated at 12–65% in the ones who have not yet experienced an incidental fall, and in 29–92% of those within the same group who have recently sustained a fall [14]. FoF is a phenomenon easily rendering itself to therapeutic interventions through general availability of effective prevention, modifications introduced to individual daily agenda, or a target-oriented rehabilitation regimen [4,14,17,18,19]. 

The present study aims to present the validation and translation procedure of the Falls Efficacy Scale—International (FES-I) tool, a scale which effectively assesses the fear of falling issue among older adults, with a view to allowing the teams of physicians, physiotherapists, psychologists, and/or psychiatrists to produce a jointly determined diagnosis, and subsequently jointly implement a specifically structured treatment management scheme. 

## 2. Methods and Materials

### 2.1. Translation of the Tool [Translation Process]

The FES-I and FES-I (Short) tool was translated in line with the translation protocol established by the members of ProFaNE [20]. Following the Linguistic Validation Manual for Patient-Reported Outcomes (PRO) Instruments, as described in the Linguistic Validation Manual, the Polish research team embarked on a linguistic validation process [21]. The Polish translation process is presented in Table 1 (further below is a summary of the tool translation): 

The linguistic validation process consists of seven steps [Table 1]—Step 1 is the conceptual definition whose aim is to clarify the concepts under study by each item of the original tool to be correctly understood and assessed in the target language. The conceptualisation of the definition is done in cooperation with the author of the tool. 

Step 2 is a preliminary adaptation of the tool translated into the cultural and language context of the target country. Step 3 consists of the process of re-translation of the Polish version of the test into the original language (backward translation) of only those items which, in the process of preliminary adaptation, should be subject to re-translation evaluation. If no item has been modified, this step may be skipped. Step 4 is a pilot test, whose purpose consists of assessing the effectiveness of the tool by persons representing the group, which is to ultimately make use of the questionnaire for diagnostic purposes in the future. 

Step 5 consists in verifying the international harmonization of the tool. This step is only applicable if more than one language version of the test is validated simultaneously. Its aim is to check the overall consistency of the translation, as well as the actual admissibility of colloquialisms in many countries, so as to ensure inter-cultural comparability of the test. Step 6 consists of proofreading, that is, checking the tool for grammatical or spelling errors. Step 7, the concluding one, is a linguistic validation report which should comprise pertinent information on the completed scope of work, the methodology applied, translation problems that may have arisen throughout, and specific decisions made in the language validation process [21]. [Figure 1].

### 2.2. The Rating Process

The FES-I questionnaire consists of 16 questions on such activities as cleaning, getting dressed and undressed, preparing simple meals, taking a bath or a shower, sitting down and getting up from a chair, climbing or descending the stairs, reaching out for something overhead or down on the ground, getting to the phone before it stops ringing, going out shopping, walking around in the neighbourhood, walking on a slippery surface (e.g., walking on an uneven surfaces (e.g., wet or icy), walking on an uneven pavement, rocky terrain, walking up or down steep terrain, visiting friends or relatives, walking in a crowded area, or going out on various occasions (e.g., a service, family meeting). The shortened version of the FES-I scale contains 7 items originating from the original FES-I questionnaire (dressing and undressing, taking a bath or shower, sitting down and getting up from a chair, walking up and down the stairs, reaching out for something overhead, or down on the ground, walking on a steep surface and going out on various occasions); it was created for practical purposes, allowing for a faster survey among the elderly individuals [14].

A sum of points ranging from 16 (no fear) to 64 (strong fear) is required to assess the outcome of the study. In the absence of more than 4 responses, the survey shall be deemed void, and in other cases, the mean value of the responses shall be calculated, multiplied by 16. The final result shall be rounded up to the nearest unsigned integer [14]. 

### 2.3. Data Collection

The research material was collected with the help of interviewers, that is, students of the Faculty of Health Sciences, Jagiellonian University of Krakow. The students received a pre-formatted database matrix in MS Excel, were instructed on the actual specifics of data acquisition/collection, on the principles of research ethics, and on the statutory requirement of securing individual informed consent from every respondent for taking part in the validation study protocol. The research was voluntary and anonymous. The study was conducted on the same study group, at two time points. 

### 2.4. Respondents

The study group consisted of a representative cross-section of seniors, residents of different regions of the country. The questionnaires were completed in the presence of a trained interviewer, following the sign-off of an informed consent form to participate in the study protocol, in conformity with pertinent recommendations of the Personal Data Protection Act. Apart from a fully voluntary character of the assignment itself, the inclusion criteria for the study group comprised the respondents’ age of 60 years and over, a good understanding of the questionnaire’s contents, and an ability to produce all required responses on one’s own. No upper age limit was set. The study protocol exclusion criteria disallowed participation in the protocol when affected by a mental disorder, as well as current medical status (e.g., hospitalisation following an accident) possibly implicating a situational bias in cognitive assessment. The study group occasionally included the interviewers’ relatives, acquaintances, close neighbours from the place of residence, as well as the inpatients of various health care facilities. A brief questionnaire on individual demographic data (i.e., gender, age, education, place of residence) was also included on the survey sheet. The second survey was conducted one week apart.

### 2.5. Statistical Analyses

The PS IMAGO PRO 6 software package (IBM SPSS Statistics 26) was applied to process the results. In order to have the research tool’s repeatability variable assessed, the respondents were asked to complete the FES-I questionnaire twice, a week apart. In the analysis of repeatability, the Wilcoxon Test was used to establish whether there were any differences between the responses provided at two time points, and the Spearman correlation analysis was also completed between the responses obtained from the first and second surveys. The Spearman rank correlation was also used to assess the tool’s accuracy, and the correlation of the FES-I results was tested against the STAI tool. Cronbach’s Alpha coefficient was calculated—the most widely used coefficient in psychology for the reliability of questionnaires, construed as the inherent consistency of the research tools. The level of statistical significance was set at *p* < 0.05.

## 3. Results

The study involved 815 respondents (N = 815). Following deletion of incomplete studies, a group of 740 respondents was statistically assessed (N = 740), F = 463, M = 277 age = 72.4 y [Table 2]. 

The validation study addressed the variables of reliability (N = 741), repeatability (N = 231) and accuracy (N = 328) [Table 3]. 

Validation tests conducted in a Polish setting showed high reliability of the research tool at issue (Alfa Cronbach = 0.932). In the psychometric tests, any scores above 0.7 were assumed to indicate satisfactory reliability of the scale; such a score had actually been obtained.

The Wilcoxon test showed no differences for the sum of the points from the individual FES-I scale questions, except for the questions: “getting dressed or undressed”, “taking a bath or a shower” and “walking on uneven surfaces, e.g., uneven pavement, rocky terrain”. In the case of FES-I (Short), the Wilcoxon test showed no differences with regard to any questions comprised in the survey questionnaire. The repeatability of the tool was thus corroborated. Correlations for the FES tool were also checked between the responses obtained initially, and one week apart. The result was r = 0.94 (*p* < 0.0001), thereby attesting to a very strong correlation.

In order to assess the tool’s accuracy, the correlation of the FES-I results was tested against the STAI tool, in Wrześniewski’s adaptation [22]. By adapting the test to the standards of the STAI tool, the number of assessed cases in the group under study was limited to the age range of 60–79 years. The Spearman’s rank correlation coefficient between the FES-I scale and STAI STAN was 0.34 (*p* < 0.001) for FES-I (N = 176), and 0.37 (*p* < 0.001) for FES-I (Short) (N = 137), which indicates a positive correlation between both tools. In both variants of the questionnaire, the correlations of FES-I results with the STAI STAN results are stronger than FES-I results with STAI CECHA results, which seems to be consistent with the perception of fear of falling, as being subject to situational constraints (characteristics).

## 4. Discussion

The Polish research team successfully completed the process of translating the tool and validating FES-I and FES-I (Short). The research method complied with the constraints of high reliability, repeatability, and accuracy. To assess the reliability of the FES-I scale, Alfa Cronbach (N = 740) was calculated (0.932), effectively attesting to the high reliability of the tool. For the evaluation of the repeatability variable, two assessments were carried out, one week apart (N = 231), making use of the Wilcoxon Test. It indicated no differences for the sum of points from individual FES-I questions, except for the questions: “getting dressed or undressed”, “taking a bath or a shower” and “walking on uneven surfaces, e.g., uneven pavement, rocky ground”. In the case of FES-I (Short), the Wilcoxon Test indicated no differences for all questions in the questionnaire, thus corroborating the tool’s repeatability. Correlation of the FES-I results with the STAI tool, as adapted by Wrześniewski [22], was tested to assess the tool’s accuracy (N = 313). Spearman’s rank correlation coefficient between FES-I and STAI STAN was 0.34 (*p* < 0.001) and 0.37 for FES-I (Short) attesting to a positive correlation between both tools. 

The FES-I tool takes into account all recommendations pertaining to the assessment of fall of falling, such as how the assessment refers to specific and defined activities, takes into account activities of various difficulty levels performed both at home and outdoors, addresses the social aspect of the fear of falling, offers a simple and clear format of responses, and also allows for a comparison of the results within a broad community of researchers [14,23,24].

The results of the validation indicate the usefulness of the tool in assessing fear of falling in older adults, both in clinical and non-clinical settings, in view of the easy practicality of the procedure. FES-I and FES-I (Short) may be applied by medical professionals, physicians, physiotherapists, psychologists and/or psychiatrists in a uniform diagnostic assessment, whilst addressing a preventive aspect, and enhancing overall effectiveness of treatment management in the elderly patients. By gaining specific insights into the physical and psychological causes of impaired and reduced individual capabilities, it may help determine and apply an adequately tailored scope of remedial measures (e.g., rehabilitation), as well as overall treatment management of existing disorders [25,26,27].

The results of the Polish language version of the scale are comparable to those obtained in other culturally modified versions of this tool. For example, a German study with 704 respondents obtained a high internal consistency coefficient (Cronbach’s α = 0.96), as well as a stability/repeatability index (r = 0.96) [28]. The Dutch version of the scale scored equally high (Cronbach’s α = 0.96, coefficient of constancy = 0.82), with a study population of n = 213 individuals and respondents aged over 70 years [29]. Similarly, in a validation study of the Short FES-I conducted on a group of 519 individuals aged 65 years and older in Japan (Cronbach’s α = 0.87), in all respective cultural versions, FES-I scale scores were significantly higher for the participants with a history of falls than among those with no previous incidents sustained [30]. 

The above-referenced examples prompted us to conclude there were manifest similarities in the results of the validation studies already completed, as well as that the research tool at issue had proven its overall effectiveness beyond reasonable doubt.

Degenerative changes, concomitant diseases, and a history of falls among older adults over 65 years of age appreciably contribute to overall reduction of individual activity and self-reliance, as well as an increase in the fear of falling [31,32,33]. Along with the rising number of seniors requiring appropriate assistance and specialist care, overall significance of the injuries sustained through falls is on the rise, as they may well result in disability and appreciably erode one’s overall quality of life [31,34]. 

## 5. Conclusions

Both FES-I and Short FES-I demonstrated adequate/excellent internal validity and reproducibility. The Short FES-I was found to be comparable to the FES-I, and may therefore be used interchangeably in research, diagnostic procedures, as well as when working along with older adults affected by a fear of falling, within the Polish population.

## Figures and Tables

**Figure 1 ijerph-19-16907-f001:**
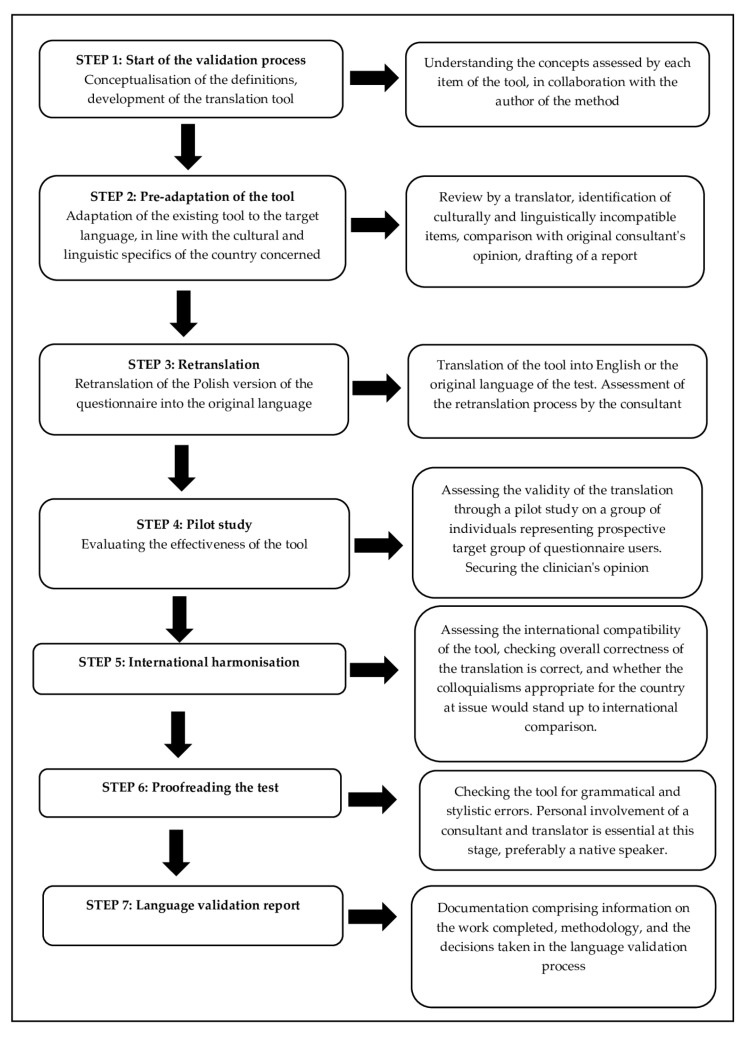
Diagram showing the linguistic validation process for the Polish cultural context, whilst making use of the procedures outlined in the Linguistic Validation Manual for Patient-Reported Outcomes (PRO) Instruments, as published by the MAPI Institute.

**Table 1 ijerph-19-16907-t001:** Table addressing the Polish linguistic validation process and the international adaptation of the FES-I tool with the aid of MAPI Validation for Clinical Trial Translation conditions.

1	Conceptualisation of the research tool	Understanding the concepts analysed by each item of the research tool, in collaboration with the author of the method.
2	Pre-adaptation of the research tool	Adaptation of the translated research tool to the cultural and linguistic specifics of the country.
3	Retranslation	Retranslation of the Polish version of the test into the original language.
4	Pilot study	Evaluation of the research tool’s effectiveness
5	International compliance	Checking the international compatibility of the research tool, if more than one language version of the test is being validated.
6	Proofreading the text	Checking the research tool for grammatical or spelling errors.
7	Language validation report	Documentation comprising information on the work completed, methodology applied, and information on the decisions taken in the language validation process.

**Table 2 ijerph-19-16907-t002:** Stratification by gender and mean age of the study subjects.

	N of Valid Ones	Mean Age	Standard Deviation (SD)	Median	Minimum	Maximum
Total	740	72.4	8.0	71.0	60.0	95.0
Women	463	72.4	8.1	72.0	60.0	95.0
Men	277	72.3	7.8	71.0	60.0	92.0

**Table 3 ijerph-19-16907-t003:** Repeatability check, respondents’ responses one week apart.

	FES 1			FES 2			p
	X	SD	Me	X	SD	Me	
Cleaning the house (e.g., sweep, vacuum, dust)	1.71	0.85	1.00	1.72	0.85	2.00	0.9388
Getting dressed or undressed	1.51	0.73	1.00	1.61	0.81	1.00	0.0040
Preparing simple meals	1.32	0.61	1.00	1.31	0.58	1.00	0.7145
Taking a bath or shower	2.17	1.07	2.00	2.28	1.21	2.00	0.0467
Going to the shop	1.72	0.88	1.00	1.74	0.91	1.00	0.9703
Getting in or out of a chair	1.75	0.89	1.00	1.79	0.89	2.00	0.3022
Going up or down stairs	2.36	1.00	2.00	2.43	1.00	2.00	0.0808
Walking around in the neighbourhood	1.74	0.87	2.00	1.78	0.91	2.00	0.4142
Reaching for something above your head or on the ground	2.18	0.98	2.00	2.21	0.99	2.00	0.9116
Going to answer the telephone before it stops ringing	2.02	0.99	2.00	2.01	0.96	2.00	0.8102
Walking on a slippery surface (e.g., wet or icy)	3.11	0.89	3.00	3.09	0.89	3.00	0.2971
Visiting a friend or relative	1.62	0.83	1.00	1.65	0.88	1.00	0.4946
Walking in a place with crowds	2.14	0.96	2.00	2.18	0.99	2.00	0.7125
Walking on an uneven surface (e.g., rocky ground, poorly maintained pavement)	2.66	0.99	3.00	2.79	0.99	3.00	0.0101
Walking up or down a slope	2.90	0.96	3.00	2.94	0.94	3.00	0.5619
Going out to a social event (e.g., religious service, family gathering, or club meeting)	1.66	0.85	1.00	1.66	0.87	1.00	0.6120
FES TOTAL	32.54	10.31	31.00	33.17	10.43	32.00	0.1762

FES 1—FES score, FES 2—FES score a week apart, p—Wilcoxon test score, X—mean, SD—standard deviation, Me—median.

## Data Availability

The datasets generated and/or analysed during the current study are available from the Corresponding Author upon reasonable request.

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
