# Peer review of "Validation of FES-I and Short FES-I Scales in the Polish Setting as the Research Tools of Choice to Identify the Fear of Falling in Older Adults"

_ijerph, 2022, doi:10.3390/ijerph192416907_

Round 1

Reviewer 1 Report

Dear Authors,

Validation studies are hard to conduct and write. The article is well written however there are some points that should be corrected:

1. Methods section: Statistical analyses is lacking. It should be written seperately. The authors wrote in results section and it is confusing the results.

2. Results: Results of comprasion of the ROC curves of FES and STAI tool may be valuable. 

3. Discussion section: Other cultural validations should be cited.  

4. Some spelling mistakes should be corrected.

Best regards.

Author Response

Authors' responses to the Reviewers

Title: Validation of FES-I and Short FES-I scales in the Polish setting as the research tools of choice to identify the fear of falling in older adults

Journal: International Journal of Environmental Research and Public Health

Special Issue:Physiotherapy and Rehabilitation as Modern-Day Medical Challenge in Older Adults

Submission ID:ijerph-2053328

The Authors have diligently addressed all the concerns raised by the Reviewers. Hopefully, the revised version of the manuscript will merit the Reviewers' satisfaction.

AUTHORS' RESPONSES TO THE REVIEWER'S RECOMMENDATIONS

REVIEWER # 1

Comments and Suggestions for Authors

Dear Authors,

Validation studies are hard to conduct and write. The article is well written however there are some points that should be corrected:

  1. Methods section: Statistical analyses is lacking. It should be written seperately.

The authors wrote in results section and it is confusing the results.

AUTHORS’ RESPONSE:

MS amended accordingly (page: 5, lines: 142-153).

  1. Results: Results of comprasion of the ROC curves of FES and STAI tool may be valuable. 

AUTHORS’ RESPONSE:

A comparison of the FES and STAI tools was made with the aid of Spearman correlation. The manuscript has been amended accordingly(page: 5, lines: 149-153).

  1. Discussion section: Other cultural validations should be cited.  

AUTHORS’ RESPONSE:

MS amended accordingly (page: 7, lines: 221-233).

  1. Some spelling mistakes should be corrected.

AUTHORS’ RESPONSE:

The entire MS has thoroughly been flushed out for any outstanding spelling mistakes or typos.

Reviewer 2 Report

The methodology for translation process is not described in detail. The authors only mention the phases of the translation process without specifying who participates in each phase (subjects, profiles of those subjects, etc.), what tools are used to develop the methodology in that phase, and how the results are analyzed.

Rating process: The authors present the questions of the questionnaire and how they are scored but do not show methods to decide the criterion validity of the score obtained. That is, the ability of the instrument (scored with the scoring system that they propose) to predict results in the dependent variable that it is supposed to be measuring.   The authors do not present the method used to demonstrate that a score on the FES-1 and FES-I is related to a given fear to falling value. Data collection- the authors explains who collects the data but not how the participants are selected and details of the people who participate in the study. In addition, it would be appreciated that they included the pre-formatted database matrix used by the evaluators   Statistical análisis- the authors do not present the statistical analyzes used The authors mention that the validation study assesses variables of reliability, replicability, and adequacy, but they do not detail how these analyzes are done and how these three result criteria are represented.

The title of table 3 should be better described as well as other information that helps to understand the results that are being presented.

Author Response

Authors' responses to the Reviewers

Title: Validation of FES-I and Short FES-I scales in the Polish setting as the research tools of choice to identify the fear of falling in older adults

Journal: International Journal of Environmental Research and Public Health

Special Issue:Physiotherapy and Rehabilitation as Modern-Day Medical Challenge in Older Adults

Submission ID:ijerph-2053328

The Authors have diligently addressed all the concerns raised by the Reviewers. Hopefully, the revised version of the manuscript will merit the Reviewers' satisfaction.

AUTHORS' RESPONSES TO THE REVIEWER'S RECOMMENDATIONS

REVIEWER # 2

Comments and Suggestions for Authors

The methodology for translation process is not described in detail. The authors only mention the phases of the translation process without specifying who participates in each phase (subjects, profiles of those subjects, etc.), what tools are used to develop the methodology in that phase, and how the results are analyzed. 

AUTHORS’ RESPONSE:

MS amended accordingly (page: 5 lines: 127-131).

The Reviewer should bear in mind that the Authors were effectively constrained by a number of specific stipulations as to how this procedure should be carried out, to the effect that no deviations whatsoever were permitted from implementing the pre-set protocol, as originally devised for this purpose by Prof. Chris Todd, University of Manchester, UK.

The Authors had to focus therefore on maintaining strict procedural conformity throughout their implementation of all applicable guidelines, with a view to securing endorsement for this validation process.

Rating process: 

The authors present the questions of the questionnaire and how they are scored but do not show methods to decide the criterion validity of the score obtained. That is, the ability of the instrument (scored with the scoring system that they propose) to predict results in the dependent variable that it is supposed to be measuring.  

AUTHORS’ RESPONSE:

The Authors would like the Reviewer to note that they are merely focused on implementing the specific requirements of the validation procedure itself, whilst following strictly all applicable formal constraints, with a view to securing official endorsement of this research endeavour, i.e. validation of their translation of the above-referenced questionnaire.

The Reviewer’s methodological concern is therefore best directed to the investigator of notable expertise in this particular domain – Prof. Chris TODD, University of Manchester:

https://sites.manchester.ac.uk/fes-i/

https://www.research.manchester.ac.uk/portal/chris.todd.html

The authors do not present the method used to demonstrate that a score on the FES-1 and FES-I is related to a given fear to falling value. 

Data collection - the authors explains who collects the data but not how the participants are selected and details of the people who participate in the study.

AUTHORS’ RESPONSE:

MS amended accordingly (page: 5, lines: 131-141).

In addition, it would be appreciated that they included the pre-formatted database matrix used by the evaluators.   

Statistical analysis - the authors do not present the statistical analyzes used.

AUTHORS’ RESPONSE:

MS amended accordingly (page: 5, lines: 142-153).

The authors mention that the validation study assesses variables of reliability, replicability, and adequacy, but they do not detail how these analyzes are done and how these three result criteria are represented.

AUTHORS’ RESPONSE:

Cronbach’s alpha reliability coefficient has been supplemented in line 163 in the MS.

The title of table 3 should be better described as well as other information that helps to understand the results that are being presented.

AUTHORS’ RESPONSE:

The title of TABLE 3 (page: 6, lines: 167) has been modified and pertinent explanations supplemented accordingly (page: 6, lines: 168-169).

Round 2

Reviewer 2 Report

Relevant improvements have been included with respect to the requested points. And/or explained some of the gaps initially detected